# Intervention Mapping for Refining a Sport-Based Public Health Intervention in Rural Schools

**DOI:** 10.3390/ijerph21121557

**Published:** 2024-11-25

**Authors:** McKenna G. Major, Janette M. Watkins, Janelle M. Goss, Derek W. Craig, Zack Waggoner, Vanessa M. Martinez Kercher, Kyle A. Kercher

**Affiliations:** 1School of Medicine, Indiana University, Bloomington, IN 47408, USA; mckmajor@iu.edu; 2Department of Kinesiology, School of Public Health-Bloomington, Indiana University, Bloomington, IN 47405, USA; janhynes@iu.edu (J.M.W.); jangoss@iu.edu (J.M.G.); 3Program in Neuroscience, College of Arts and Sciences, Indiana University, Bloomington, IN 47405, USA; 4Department of Health Promotion and Behavioral Sciences, UTHealth Houston School of Public Health, Houston, TX 77030, USA; derek.w.craig@uth.tmc.edu; 5White River Valley Middle School, Lyons, IN 47443, USA; zwaggoner@wrv.k12.in.us; 6Department of Health & Wellness Design, School of Public Health-Bloomington, Indiana University, Bloomington, IN 47405, USA; vkercher@iu.edu

**Keywords:** implementation science, sport-based youth development, intervention, children

## Abstract

Sport-based youth development programs, delivered through campus–community partnerships, can create impactful experiences for college students, meet university objectives, and improve the health of children in under-resourced rural communities. This study aimed to pilot test intervention mapping (IM), a systematic approach to intervention development and implementation, to refine the Hoosier Sport intervention, which is a local public health initiative that utilizes the Obesity-Related Behavioral Intervention Trials (ORBITs) model to improve physical activity in middle school children. The IM process, which included a diverse IM planning and advisory group of university representatives and local schools, was guided by self-determination theory (SDT) and social cognitive theory (SCT) and followed four steps: Logic Model of the Problem, Logic Model of Change, Program Design, and Program Production. Using SDT and SCT, we identified our personal determinants as autonomy, competence, and relatedness, while our environmental determinants were role-modeling and sports equipment access. We then created change methods and practical applications for refining and implementing our intervention and gathered pilot test data to assess the feasibility of the intervention. The IM process provided a more robust and evidence-based approach to intervention design and production, while involving stakeholders to foster meaningful collaboration and increase program success. By using IM in program development, public health interventions that promote youth development through sport will likely be more easily scaled up.

## 1. Introduction

Only 1 in 5 youth meet the United States guidelines for aerobic and muscle and bone-strengthening physical activity (PA) [1]. This lack of PA can result in poor physical, intellectual, and emotional health. Children with low levels of PA are at an increased risk of childhood obesity, cardiovascular disease, depression, and poor cognitive outcomes [2,3]. These health problems lead to decreased quality of life in children and large economic costs tied to the treatment and long-term care of these diseases [4]. Furthermore, marginalized groups like those in rural and low socioeconomic (SES) settings are disproportionately affected by distinctive barriers to PA, such as limited resources and a lack of support from parents and peers [5]. Children’s PA interventions serve as a public health approach that may be implemented to improve long-term outcomes for youth, especially those in underserved areas.

Various approaches to improving PA in children, including both curricular and extracurricular changes through school [6], home, and community settings [7], have been implemented in the previous literature. However, many of these interventions often fail to address multiple environmental levels and various settings, instead focusing on individual determinants. Intervention programs may also face limitations in the form of balancing application and reproducibility. Some studies lack the rigor and structure of a well-controlled study or the pragmaticism and translation seen in observational studies. Furthermore, many PA interventions may also fail to scale up or scale up prematurely in the design and implementation process, as they do not gather data on the feasibility of the intervention [8]. Due to the limitations of existing evidence, there is no definitive conclusion on the sustainability or scalability of PA interventions on children.

Although there is limited evidence of the feasibility of youth PA interventions, this does not necessarily indicate that such interventions are not effective in increasing youth PA, but rather, more research must be conducted involving interventions that are both rigorous and able to be translated to real populations. We argue that the majority of current studies on youth PA interventions lack system development, theory-based evidence, and a multilevel approach that emphasizes both the individual and community [9,10,11]. In order to create an intervention with greater practicality and impact, it is important to reach a balance between a well-controlled study with rigor and reproducibility and an intervention that will be applicable to real-world settings at multiple levels. However, it can be difficult to strike this balance using many of the current approaches to PA interventions.

One possible approach in creating effective and feasible interventions is through a systematic procedure of intervention development, such as intervention mapping (IM). IM is a systematic approach to the development and implementation of health-promoting interventions and has been found to be an efficacious strategy for improving children’s health [12,13,14,15]. During the six-step IM process, researchers work with community stakeholders to identify and understand the health problem, specify targets of the intervention, design intervention materials, and plan for implementation and evaluation. Decision-making within each step of the IM process is iterative as well as theory- and evidence-informed to ensure a detailed and effective intervention. Furthermore, IM utilizes an ecological approach to intervention development which guides users in the design of multi-level interventions to intervene at the individual, organizational, and community levels. The IM presents a systematic process for designing and implementing public health interventions.

A key characteristic of the IM process is the integration of evidence-based behavioral theory. An essential part of the IM process is using empirical evidence and theories to understand the behavioral and environmental factors leading to the health problem. Once these factors are identified, program design is a step that considers theoretical methods for modifying determinants and creating performance objectives. The IM framework is also a flexible design process that recognizes that one or several behavioral theories may be necessary to understand and outline determinants and performance objectives. For PA interventions in children, several behavioral theories may be necessary to understand what factors affect children’s health and activity levels and how these factors can be modified. Intervention programming grounded in theory explains why researchers investigate specific determinants and change methods. Additionally, identifying theories used in intervention development allows researchers to make future adaptations to the program.

In the present study, we describe the development and retrospective integration of a sport-based physical activity intervention for rural middle school children in the Midwestern United States using IM. Hoosier Sport exists within Phase I of the Obesity-Related Behavioral Intervention Trials (ORBITs) model [16]. The primary objective of the study was to use steps 1–4 of IM to identify the (a) Logic Model of the Problem and Change; (b) behavioral and environmental determinants; (c) performance objectives and change methods; and (d) the conceptual/theoretical framework for the Hoosier Sport intervention. The secondary objective was to synthesize the IM results within the context of two initial pilot studies of Hoosier Sport in rural school children.

## 2. Materials and Methods

### 2.1. Overall Study Design

This is a mixed-methods study describing the completion of IM steps 1–4 as part of the process to develop and optimize the Hoosier Sport intervention. For Hoosier Sport, this paper reports the IM tasks completed as part of the broader development project. In this larger project, we established a planning team comprising four research team members, and an advisory panel consisting of two school leaders where the intervention would take place. Intervention mapping was completed retrospectively for refinement of the program, and further pilot testing.

### 2.2. Research Context

#### Hoosier Sport

Hoosier Sport is a multi-component sport-based youth development program that develops campus–community partnerships between a Midwestern university and under-resourced rural school districts. Hoosier Sport begins with a rural school district needs assessment survey, followed by a human-centered participatory co-design process to develop the intervention, and pilot testing. Thus far, the intervention has included combinations of enhanced sport-based physical education classes (provided by trained college students), goal setting, in-class access to sports equipment, and physical activity incentive systems. Hoosier Sport uses a Type 3 Hybrid study design that allows for an assessment of feasibility and clinical outcomes simultaneously. The program is within Phase 1 of the ORBIT model. The ORBIT model is a program development framework emphasizing the recognition of behavioral and environmental determinants and the continuous refinement and testing of the program design [16]. Primary outcomes include (a) implementation outcomes (e.g., trial- and intervention-related feasibility indicators) and (b) various PA measures (e.g., daily moderate-to-vigorous physical activity [MVPA] and total daily PA [including all intensity levels]). Secondary outcomes include psychological outcomes (e.g., basic psychological needs, self-perception profiles) and leadership skills (e.g., goal setting). Hoosier Sport can be applied in physical education settings, before school and/or after school. The delivery model for Hoosier Sport consists of training college service-learning students to deliver interventions through a university-approved course curriculum. The students were trained at a 5 h pre-school year team training event and at recurring bi-weekly team meetings leading up to the intervention start date. Beginning in 2024, college students will be trained in the university-approved service-learning course prior to being deployed in the community.

### 2.3. Planning Team Eligibility and Recruitment

Planning group: Members of the Hoosier Sport research team, including undergraduate, graduate, and medical school students, allied health practitioners, and faculty researchers. We conducted a total of 11 in-person meetings over the course of 3 months at a research-based university in the Midwestern United States. Planning team members were invited to meetings when the mapping tasks and agenda were directly relevant to them. The research team members participated in all intervention mapping tasks.Advisory Panel: academic researchers from a Midwestern university, college students, partner school district members (i.e., teachers, administrators, parents).

### 2.4. Theories, Models, and Frameworks for Implementation Strategies

#### 2.4.1. Self-Determination Theory

The basic psychological needs mini-theory (BPNT) within the self-determination theory (SDT) outlines three human needs of autonomy, competence, and relatedness that when met by an individual, lead to great self-motivation and health [17]. We selected this theory to guide our selection of determinants and creation of change matrices due to the theory’s explanation of psychological factors as a background for personal behaviors. Following this theory to design and detail our intervention, children who attain higher levels of autonomy, competence, and relatedness in our programming are more likely to achieve a greater well-being.

#### 2.4.2. Social Cognitive Theory

The social cognitive theory (SCT) explains that there are three reciprocal contributions made by personal, behavioral, and environmental domains [18]. This framework also highlights cognitive processes such as self-efficacy, goals, and outcomes that impact and are impacted by these domains. We utilized this theory in creating our determinants and change matrices due to the emphasis it placed on environmental factors, such as social norms and access to resources, to explain behaviors and cognitive processes. The theory posits that environmental factors influence behavioral and personal factors, which can be used to address such environmental determinants in intervention design.

### 2.5. Intervention Mapping Tasks

#### 2.5.1. Step 1: Needs Assessment, Logic Model of Problem

The first step of IM consists of the research team identifying and describing the problem that the intervention will address. The Hoosier Sport program was created to address cardiovascular disease (CVD) in a rural, under-resourced youth population. Heart disease is the leading cause of death in Indiana [19], and rural populations are at a greater risk for CVD [20], in part due to increased barriers to PA [5]. A lack of PA is a risk factor for not only CVD but also diabetes, decreased mental health, cognitive decline, and musculoskeletal injuries [2,3,21]. PA in rural, low SES populations can often be inaccessible due to financial, social, and structural barriers [22]. A previously published needs assessment for our priority population was utilized to specifically address the barriers and needs of the community [23]. Our planning group leveraged the previously collected needs assessment data to populate the Logic Model of the Problem.

#### 2.5.2. Step 2: Logic Model of Change and Performance Objectives

Step 2 is centered on establishing performance objectives to address personal and environmental determinants. We first determined our environmental and behavioral outcomes for the at-risk population and stated the performance objectives tied to each outcome. After reviewing the current literature, we selected determinants based on SCT and BPNT. Utilizing both our determinants and performance objectives, we created change matrices for guiding potential change methods for use in Step 3.

#### 2.5.3. Step 3: Program Design

For Step 3, we further designed our program by planning the theme, components, and scope of our intervention. This step also included selecting change methods that are then translated into practical applications. Change methods are theory-based processes for creating change in our determinants from Steps 1 and 2. Practical applications are the specific approaches and procedures put into place to deliver our change methods. We selected our change methods and practical applications based on the SCT and SDT theories, feasibility outcomes from previously published pilot testing [24,25], and guidance from our community partner.

#### 2.5.4. Step 4: Program Production

As part of Step 4, we analyzed the existing materials and curriculum, while considering the cultural relevance of the context of the intervention. Hoosier Sport already had materials and a protocol, but the IM process helped improve and refine the program’s design. One unique aspect of the Hoosier Sport program is its delivery method, which relies on college students in a service-learning course. The college students help to facilitate the program activities, while serving as role models for the children.

Furthermore, in addressing the potential cultural issues within the program, we discussed mechanisms for increasing cultural relevance. Due to the distinct values and norms within a rural population and economic disparities that may exist in our target population, we involved our community stakeholders throughout the program development and IM process. The insights of our community partner help to ensure that our program aligns well with the resources, values, and lifestyles of the population.

After drafting our program, we gathered pretest and pilot data to determine the effectiveness and feasibility of our intervention [24,25].

### 2.6. Power Analysis and Data Analysis

Data analysis was conducted using the latest version of R Studio (2023.12.0 + 369), and figures were generated using the most recent version of MATLAB (R2022b).

## 3. Results

### 3.1. Intervention Mapping Tasks

#### 3.1.1. Step 1: Logic Model of the Problem

For Step 1, we utilized SDT and SCT to identify our behavioral determinants as autonomy, competence, and relatedness, and our environmental determinants as role-modeling and sports equipment access [17,18]. The environmental conditions were then identified at the interpersonal (teachers, college students, peers, and reinforcement), organizational (lack of trained staff, PE policy, free play opportunities, and sport infrastructure/equipment), and community level (family beliefs). Combining our determinants, environmental conditions, and the health problem, we created a Logic Model of the Problem (Figure 1).

The overall goal of the intervention was to significantly lower children’s resting heart rates from pre- to post- intervention, as the resting heart rate has been utilized in the previous literature as a marker for cardiovascular health [26]. This was supported by a behavioral specific goal of encouraging children to meet daily step goals, which provides an understandable and relevant goal that is tied to our determinants of autonomy, competence, and relatedness. Our environmental goal centered around promoting role-modeling and providing accessible sports equipment.

#### 3.1.2. Step 2: Logic Model of Change

For the next step, we outlined our behavioral health outcomes as increased total PA by 0.5 standard deviations (SDs) and meeting a step goal 5/7 days a week, as well as environmental outcomes such as improving accessibility of sports equipment and promoting role-modeling. Increased PA and step goals were chosen due to the association of both with a decreased CVD risk [27,28]. Furthermore, increasing accessibility to sports equipment has also been found to improve youth PA levels [29], and role-modeling from college students is a new and untested approach to interventions. For each outcome, we combined performance objectives, which are defined as the necessary steps to reach the outcome, and our determinants from Figure 1 to create change matrices (Table 1). The determinants, performance objectives, and change methods were then synthesized to create the Logic Model of Change (Figure 2).

#### 3.1.3. Step 3: Program Design

In Step 3, we developed the curriculum of Hoosier Sport according to the change matrices in Table 1. These activities were iteratively refined to align with the identified determinants using theory- and evidence-based change methods. SDT, BPNT, and SCT provide the foundation for understanding behavior change. BPNT emphasizes intrinsic motivation, which is more sustainable than extrinsic motivation. It posits that satisfying basic psychological needs—autonomy, relatedness, and competence—is fundamental for well-being and drives motivation. Unstructured and structured physical activities can satisfy these needs, promoting sustained engagement in sports and overall well-being (Figure 3). The Hoosier Sport program provides opportunities for both structured at school activities, such as sports and drills, and unstructured at home PA opportunities, like meeting incentivized step goals, thus supporting the satisfaction of basic psychological needs.

Additionally, activities involve choices in sport activities (e.g., choosing between drills and teammates) to support the autonomy of students and promote relatedness among peers. Also, the increase in sport skills that accompanies an extended program like Hoosier Sport also fosters greater feelings of competence. SCT identifies key factors like self-efficacy, role modeling, and social support, which influence the initiation and maintenance of health-promoting behaviors. By using role models (e.g., college service-learning students) and social support for sport participation, Hoosier Sport helps to create lifelong sport participation behaviors in children (Figure 4).

Importantly, meeting basic psychological needs such as autonomy and competence helps to increase self-efficacy, further supporting the formulation and maintenance of health behaviors (Figure 5).

#### 3.1.4. Step 4: Program Production

Pilot testing is an essential aspect of IM and ensures the feasibility of an intervention in a real-world context. We underwent a systematic approach in which we collected smaller-scale pre-test data and larger-scale pilot test data. Our pre-test plan at an individual level included collecting data from children during a PE class (6-Minute Walk Test (6MWT), plank, total PA, total steps, and baseline Canadian Assessment of Physical Literacy (CAPL-2 scores)). At an environmental level, we assessed the service-learning student program with questionnaires on beliefs for the college students. Along with obtaining pre-test data, we also gathered pilot testing data for the entire intervention.

### 3.2. Pilot Test

#### 3.2.1. Pilot Test 1 Results

In the first pilot study, a total of 24 6th–7th graders were included, with 58% females and 42% males, aged 10 to 13 years (M = 11.92, SD = 0.98) [24]. A Hybrid Type 3 design was employed, enabling the examination of changes over time and the assessment of the Hoosier Sport intervention’s effects at both individual and group levels. This design allows for simultaneously assessing implementation feasibility indicators while evaluating clinical outcomes. The Hoosier Sport intervention spanned 8 weeks, organized into two 4-week sport units—soccer during weeks 1–4 and pickleball during weeks 5–8. Feasibility was assessed through recruitment capability, retention, attendance, acceptability, appropriateness, and compliance with pedometer wear and step tracking. Psychological needs were measured using the child-adapted Basic Psychological Needs in Exercise Settings (BPNES) scale. Physical activity and literacy were evaluated with the CAPL-2 and the 6MWT.

Most participants found the Hoosier Sport program likely to succeed, feasible, and engaging. Over 91% agreed that the program was exciting, and 73% felt it was a good fit for their school. Measures of autonomy, competence, and relatedness showed high initial scores and remained stable throughout the intervention, although changes were not statistically significant. Significant improvements were observed in the 6MWT, with a mean increase of 31.03 m (*p* = 0.026). No significant changes were found in plank test scores or the resting heart rate. However, significant differences in heart rate (*p* = 0.001) and systolic blood pressure (*p* = 0.019) before and after the 6MWT were observed.

#### 3.2.2. Pilot Test 2 Results

The second pilot study consisted of 35 6th–7th grade rural students (test group: *n* = 24, control group: *n* = 11), aged 10–13 (M = 12.05, SD = 0.35) [25]. The study employed a Hybrid Type 2 design to assess the implementation feasibility and clinical outcomes of the Hoosier Sport intervention over an 8-week period. The intervention included strength training sessions in weeks 1–4 and basketball activities in weeks 5–8. Data were collected at baseline (week 1), mid-intervention (week 4), and post-intervention (week 8), with feasibility metrics such as attendance and retention tracked throughout. An incentive reward system was implemented to encourage at-home fitness activities, alongside in-person strength sessions led by trained research assistants and graduate students. Participants were assessed using physiological and psychological measures, including the Basic Psychological Needs Satisfaction and Frustration Scale (BPNSFS), Self-Perception Profile for Children (SPPC), and Axivity AX3 accelerometers.

The study successfully recruited 23.3% of the total middle school enrollment, with a retention rate of 94.3%. Attendance rates were strong, with 80.15% for strength training sessions and 88.45% for basketball sessions. Feasibility indicators revealed high acceptability, appropriateness, and overall feasibility scores. The intervention demonstrated significant improvements in psychological outcomes for the test group, with increases in autonomy (mean increase = 2.78, *p* = 0.023), social competence (mean increase = 2.48, *p* = 0.005), and global self-worth (mean increase = 1.38, *p* = 0.028). Physical activity levels, measured by Axivity AX3 accelerometers, showed significant increases from pre- to post-intervention in both the control and test groups for total physical activity (test group: *p* = 0.031, control group: *p* = 0.026) and daily steps (test group: *p* = 0.025, control group: *p* = 0.041).

## 4. Discussion

Because children’s PA interventions continue to be scaled up prematurely or fail to reach dissemination and implementation phases [8,30], more robust public health approaches have become increasingly important for intervention development and evaluation [13,14,15]. The current study yielded three key findings. First, IM provided a valuable structure to refining the intervention design process. Second, IM guided the integration of evidence-based behavioral theory throughout the creation of a framework for measuring physical and psychological outcomes, as well as a curriculum that is feasible and attractive for students. Third, IM ensured that stakeholders were included in the research process to improve program feasibility and create meaningful collaboration with the community partner. Together with our previous Hoosier Sport pilot studies [24,25,31], the present study supports transparency with a practical example of the often-ambiguous intervention development process. Simultaneously, contextualizing IM within the context of our existing intervention, two pilot tests, and potential future trials provides a valuable example for public health researchers looking to align with the iterative cycle of refinement that is recommended best practice within the ORBIT model.

Our first key finding was that the IM process provided a systematic framework for the development and evaluation of the Hoosier Sport program. Through IM, we created a detailed, step-by-step protocol for our intervention that included specific methods for generating change in our target population and measurable outcomes. The clear identification of our health problem and population’s needs allowed us to assess the barriers and personal/environmental determinants of the program, while our performance objectives and change methods provided structured mechanisms for designing activities within our intervention. In identifying determinants and creating methods of change, we implemented a multi-level approach that addressed individual, organizational, and community levels. We were also able to identify and clarify the program’s outcomes and goals, which helps in program assessment and evaluation in future pilot testing. The production of a structured, multi-level protocol for our intervention helps in future translation and scaling up due to its increased internal and external validity. We designed an intervention that has strong rigor and pragmaticism, with the goal of enhancing past iterations of the Hoosier Sport program [24,25]. Similar to other studies, the results demonstrate that our program can be more easily applied to other settings and locations, due to a systematic process for designing and implementing the intervention [14,32]. Furthermore, the IM process allows for continual partner feedback as well as feasibility assessments that help to strengthen the protocol in its ability to be scaled up to larger populations. The IM process provided a systematic framework that enabled the development of a detailed, scalable intervention for the Hoosier Sport program, ensuring clear goals and broad applicability.

Our second key finding was that IM guided the integration of key behavioral theories into the design of the intervention. While some interventions incorporate a single behavioral theory to guide the planning process, our program utilized multiple theories [33]. In this study, we successfully integrated SDT and SCT into the design of our sport-based youth development program, which played a crucial role in shaping its effectiveness. For example, SDT, which emphasizes the psychological needs of autonomy, competence, and relatedness, was instrumental in guiding our program’s structure. In Pilot Test 1, we aimed to create an environment where participants could exercise choice, develop their skills, and build supportive relationships [24]. This approach was well received, with over 91% of participants finding the program exciting. Although the psychological measures of autonomy, competence, and relatedness remained stable, participants experienced significant physical improvements, such as significant improvement in the 6MWT. Similarly, SCT provided a valuable framework for understanding how personal, behavioral, and environmental factors interact to influence behavior. This was evident in Pilot Test 2, where we focused on enhancing self-efficacy through structured strength training and basketball sessions [25]. The study showed significant improvements in psychological outcomes, including increased autonomy, social competence, and global self-worth, among the test group. Additionally, both the test and control groups saw significant increases in physical activity levels such as total PA, daily steps, and the 6MWT, indicating the program’s success in promoting sustained behavior change. Together, SDT and SCT were essential in guiding the development of a comprehensive intervention that addressed both the psychological and environmental aspects of youth development through sports.

Our third key finding was that including stakeholders in the IM process increased the program’s success. A crucial factor in building trust and fostering meaningful collaboration in community-based research depends on the engagement and inclusion of stakeholders in leading roles, allowing stakeholders’ voices to be seen in the research development process [34]. For community-based research to succeed, it is essential that all involved stakeholders share a common understanding of the research program’s objective. In addition to including children in the planning process, as seen in other interventions, school faculty members were also part of the IM team [35]. Through following the IM steps, there were multiple opportunities for stakeholders of the targeted community to voice their opinions and aid in the development of Hoosier Sport. Moreover, research that claims to represent children’s voices may be flawed due to adult–child power dynamics, a lack of diverse child perspectives, and the problematic reliance on parents as proxies for children’s viewpoints [36]. By including children in the IM process, this allowed them to have a voice heard in the development of research targeted at them. In Hoosier Sport, the co-design phase of the IM process included children averaging 12.6 years in age (n = 6) and adults averaging 43.3 years in age (n = 6) [31]. Thematic analysis revealed that both children and adults valued autonomy and choice in activities, with children focusing on peer relationships and adults emphasizing relatedness through positive role modeling. Lastly, both children and adults were also included in the needs analysis process of IM (n = 39 and n = 63, respectively). From the needs analysis, it was found that the majority of children (71.8%) and the overall sample (82.2%) showed interest in new physical activity programming, with 89.7% of children and 96.8% of adults recognizing its health benefits [23]. Overall, including stakeholders in the IM process is crucial as it ensures that diverse perspectives and needs are addressed, leading to more relevant and effective interventions.

There were several limitations to the IM process and results. While IM was useful, there are important challenges to remain cognizant of. As has been acknowledged in other literature [37], intervention mapping can quickly become overly complex and jargon-filled, leading to a lack of practicality or difficulty in application and translation. As our planning group worked through IM steps 1–4, we continually ensured that we returned to definitions of key terms, because many of the terms were not immediately clear (e.g., determinants, matrices, performance objectives, etc.). Another challenge was obtaining stakeholder feedback at the right level of depth while also presenting the materials and questions in a clear and concise manner. Many community-based stakeholders care about the populations they work in, but it can be more difficult to help them see the practicality or importance of IM tables, concepts, and processes. One solution in Hoosier Sport was using the co-design process to create a space where stakeholders could engage in the IM process in a more naturalistic setting through semi-structured interviews [31], and then the research/IM team could incorporate this information into the IM process. Lastly, there is difficulty in going from intervention development to addressing implementation, but it is a critically important step. Researchers have acknowledged the importance of this challenge and have subsequently developed an entire other process stemming from the next step (step 5) of intervention mapping, called implementation mapping [38]. We will complete steps 5 and 6 in a future publication focusing on implementation mapping and program evaluation. Notwithstanding these challenges, IM was a useful tool for refining our prior iterations of Hoosier Sport into a more specific, measurable, intentional, and likely more effective evidence-based intervention for improving public health.

## 5. Conclusions

This study demonstrates the practicality and efficacy of using the IM process to design and refine an intervention that can be more easily translated and scaled up. Through IM, we created a systematic framework for a youth physical activity program, in which the curriculum and its delivery are based on evidence-based theory. We created Logic Models of the Problem and Change, utilized SDT and SCT to identify our determinants, and produced specific performance objectives and change methods for a structured, multi-level approach. Furthermore, in order to build significant collaboration with our community partner, the process also continuously incorporated stakeholders’ feedback, specifically including input from the children in the program. This feedback increased children’s acceptability of the program, as they were given a voice in program design. Through pilot testing, we found that children found the intervention to be feasible. Although the IM process can at times be considered overly complex and obscure, this study aims to bring transparency to the process in the context of our current Hoosier Sport program and pilot studies.

## Figures and Tables

**Figure 1 ijerph-21-01557-f001:**
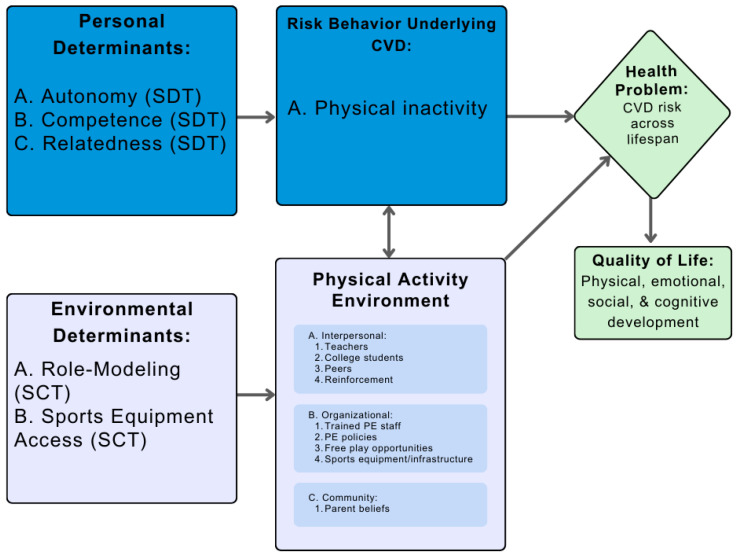
Logic Model of the Problem (SDT: self-determination theory, SCT: social cognitive theory, CVD: cardiovascular disease, PE: physical education).

**Figure 2 ijerph-21-01557-f002:**
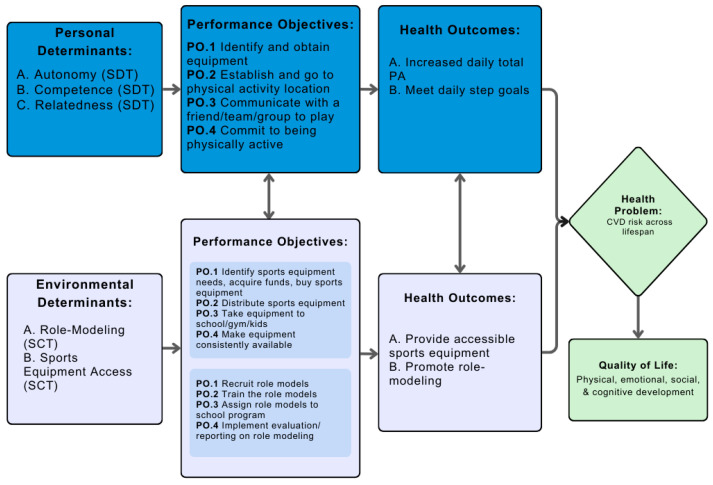
Logic Model of Change.

**Figure 3 ijerph-21-01557-f003:**
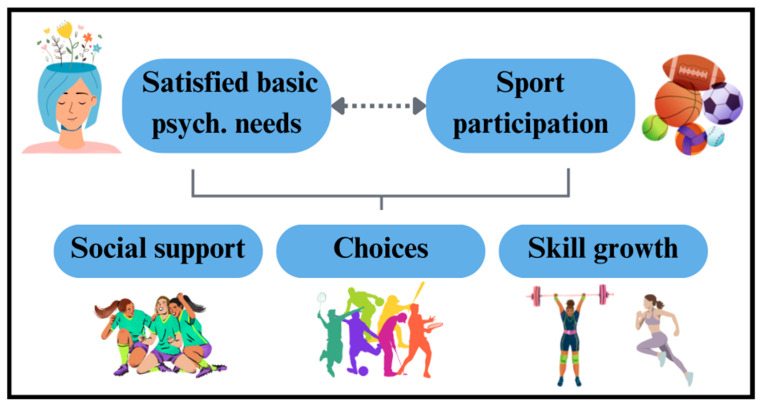
Summary of basic psychological needs theory and PA behaviors.

**Figure 4 ijerph-21-01557-f004:**
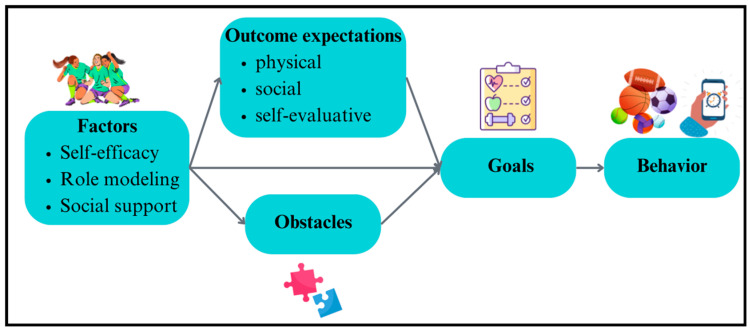
Summary of social cognitive theory and PA behaviors.

**Figure 5 ijerph-21-01557-f005:**
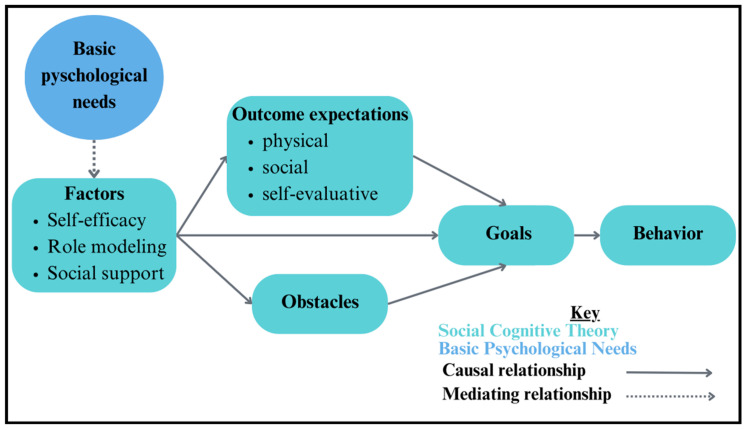
Synergy of BPNT and SCT.

**Table 1 ijerph-21-01557-t001:** Matrices of change objectives.

Behavioral Outcome: (1) Increased PA (Total PA) (2) Meeting Step Goal			
Performance Objective	Personal Determinants		
	Determinant 1: Increased autonomy to choose a PA	Determinant 2: Increased competence to perform PAs	Determinant 3: Increased relatedness to perform PA
PO.1 Identify and obtain the equipment	Obtain equipment without needing help	Display knowledge of where to obtain equipment and what equipment to obtain Establish access to the equipment (i.e., key to closet)	Demonstrate ability to choose appropriate equipment for group activity Identify amount of equipment needed for the group
PO.2 Establish and go to the physical activity location	Express choice to go to location	Identify how to get to location (i.e., directions) Display confidence in ability to get to location (i.e., riding a bike, walking far)	Communicate with friends/peers/adults for a ride Choose location that supports group play
PO.3 Communicate with friend/team/ group to play	Display motivation to be with others/peers Choose to engage in discussion	Exhibit social skills to communicate/ engage Demonstrate self-confidence in skills	Establish friends/social connections Choose which friends to play with Set groups’ goals for steps/PA
PO.4 Commit to being physically active	Display motivation to be active and meet step/PA goal Choose which type of activity to engage in	Gain physical literacy Recognize step/PA goals while being active	Agree to participate in team-based sports/activities
Environmental Outcome: Providing accessible sports equipment			
Performance Objective	Environmental Determinants		
	Determinant 1: Equipment access to choose different sports		
PO.1 Identify sports equipment needs, acquire funds, buy sports equipment	Allocate research funds for sports equipment Conduct needs assessment with community		
PO.2 Distribute sports equipment	Have access/partnership with school Identify audience/ population (i.e., PE class, free play, health class)		
PO.3 Take equipment to school/gym/kids	Have team members available to distribute equipment Have means of transportation for team members		
PO.4 Make equipment consistently available	Provide readily available equipment during school hours Ensure equipment is always available in free-play areas		
Environmental Outcome: Promoting role modeling			
Performance Objective	Environmental Determinants		
	Determinant 1: Positive role models for PA		
PO.1 Recruit role models	Use newsletters at IU Interview students for eligibility		
PO.2 Train the role models	Students complete service-learning course Research assistants complete Canva training		
PO.3 Assign role models to school program	Students complete background checks Review student goals and assign to PE or health class		
PO.4 Implement evaluation/ reporting on role modeling	Identify measures of implementation Collect data pre- vs. post- Prepare report/manuscript of results		

## Data Availability

The datasets used and/or analyzed during the current study are available from the corresponding author on reasonable request.

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
