# Peer review of "Intervention Mapping for Refining a Sport-Based Public Health Intervention in Rural Schools"

_ijerph, 2024, doi:10.3390/ijerph21121557_

Round 1

Reviewer 1 Report

Comments and Suggestions for Authors

Thank you for the opportunity to review this paper on Intervention mapping for refining a sport-based physical activity intervention in rural schools. Increasing PA for children and youth continues to be a challenge in teh U.S., so this and other similar efforts to understand the potential efficacy of interventions are critical public health steps. 

Overall, i was very pleased with the paper, including the clarity of descriptions, the detailed methodology, and the description of the different models used and considered. Therefore, most of my comments are only in regard to improving the writing, along with a few questions to consider. 

Line 23: indicates four steps are used, but those steps are not identified in the abstract? 

Line 28: I’m not sure ‘incorporates’ is the best word to use in that context.

Line 39: ‘these changes’ don’t actually reflect changes. Instead, it seems they are discussing outcomes described in the previous sentence? 

Line 41-42: disproportionately affected towards what? I believe you mean towards PA, but that is not clear. 

Line 43: philosophical question: should only decreased PA be the target for interventions, or should it be targeted to all children and youth? 

Line 11 and 118: ‘Thus Far’. Try not to use the same phrase/sentence opener twice in the same paragraph. 

Line 152: Sentence starts with ‘It’. I would suggest replacing It with what you are actually describing.

Fig 2: Logic Model. I am not familiar with this particular model, so my comment might not make sense, but the final Diamond is titled “Health Problem’, with a sub-text of ‘decreased CVD risk across the lifespan’. Is decreased CVD risk a good thing, not a ‘health problem’? 

Discussion: this is generally well written, and I particularly like the clarity in describing each of the four key findings. My question, however, is that there are no obvious limitations listed for this work, which seems an odd omission. In fact, teh fourth key finding does mention a lot of challenges, but is presented as ‘key findings’, not limitations. Were there in fact limitations that would be helpful for readers to know about?

Reviewer 2 Report

Comments and Suggestions for Authors

Title: correct.

Summary: It is advisable to enter a brief description of the sample used (number, ages...). The general outline of the summary is adequate, but we are missing this section regarding the participants.

Keywords: it is advisable to enter a keyword that refers to the type of sample used in the study (adolescents, for example).

Introduction.

Authors are advised:

- Review the concepts of “adolescence” and “childhood”. In the introduction section reference is made to childhood, an aspect that could lead us astray when focusing on the participants in this study, who are young people or adolescents, terms also used by the authors.

- Lines 55-56. This reviewer has serious doubts about this statement by the authors. The authors are asked to be clear in this regard or, on the contrary, to word this statement differently.

- Line 57-66. The authors do not provide evidence for the claims they present in this part of the study. This reviewer is confident that there is evidence on the effectiveness of physical activity interventions in young people, and the results are effective. Therefore, authors must explain and demonstrate that their claims are true.

- Lines 67-77 and 78-89. There is still a lack of references that give value and credibility to the authors' statements.

- Lines 90-98. The authors continue to incorrectly use words such as “children” to refer to the sample of this study.

Design.

- If steps 5 and 6 are to be included in future research and are not shown in this study, they cannot be established as study objectives.

Results.

- Lines 196-311. All this description does not correspond to the results of the study. Therefore, this is not the section of the study to refer to these contents, which should be located, reviewed and summarized in the methodology section of the study; even, in relation to the description of Hoosier Sport, in the theoretical section of the study (introduction). All this information is meaningless in this part of the investigation.

In this sense, the authors are required to carry out an in-depth review of this section.

- The results of pilot tests 1 and 2 would fit into this section of the study.

- Without the initial (theoretical) part, the results of the study are limited.

Discussion.

- Lines 359-374. More evidence (references) is needed to contrast the results shown in this study.

- Lines 375-395. Evidence (references) is needed to contrast the results shown in this study. The authors should keep in mind that they are developing the discussions (of the results) of their study.

- Lines 396-416. More evidence (references) is needed to contrast the results shown in this study.

- Lines 417-439. More evidence (references) is needed to contrast the results shown in this study.

- Lines 440-459. This section is more similar to a “limitations of the study and proposals” section than to a discussion of the results section. The authors must review this aspect and establish a study limitations section, with information contained in this section.

Conclusions.

The authors must respond, one by one, to the objectives established for this study (lines 90-98), in a clear and concrete manner.

References

Total: 29 references.

16 current references (55.17%) from the last 5 years.

13 references prior to 2019 (44.83%).

Round 2

Reviewer 2 Report

Comments and Suggestions for Authors

Many thanks to the authors for addressing all the proposed revisions. This reviewer considers that aspects as fundamental as introducing scientific evidence of certain aspects shown by the authors, and the fact of reorganizing the information in the manuscript taking into account its different sections, considerably improve the initial version of the same.

Despite the initial presentation being a good work, this reviewer considers that it is still a good work and improved in terms of clarity, both for the reader and for the scientific community in general.

Author Response

Comment 1: Many thanks to the authors for addressing all the proposed revisions. This reviewer considers that aspects as fundamental as introducing scientific evidence of certain aspects shown by the authors, and the fact of reorganizing the information in the manuscript taking into account its different sections, considerably improve the initial version of the same. Despite the initial presentation being a good work, this reviewer considers that it is still a good work and improved in terms of clarity, both for the reader and for the scientific community in general.

Response 1: Thank you for this comment! We appreciate you taking the time to review our manuscript.